# Emergent increase in coral thermal tolerance reduces mass bleaching under climate change

**Liam Lachs** [1,2] ✉, **Simon D. Donner** [2], **Peter J. Mumby** [3,4], **John C. Bythell** [1], **Adriana Humanes** [1], **Holly K. East** [5] & **James R. Guest**[1]

Recurrent mass bleaching events threaten the future of coral reefs. To persist under climate change, corals will need to endure progressively more intense and frequent marine heatwaves, yet it remains unknown whether their thermal tolerance can keep pace with warming. Here, we reveal an emergent increase in the thermal tolerance of coral assemblages at a rate of 0.1 °C/decade for a remote Pacific coral reef system. This led to less severe bleaching impacts than would have been predicted otherwise, indicating adaptation, acclimatisation or shifts in community structure. Using future climate projections, we show that if thermal tolerance continues to rise over the coming century at the most-likely historic rate, substantial reductions in bleaching trajectories are possible. High-frequency bleaching can be fully mitigated at some reefs under low-to-middle emissions scenarios, yet can only be delayed under high emissions scenarios. Collectively, our results indicate a potential ecological resilience to climate change, but still highlight the need for reducing carbon emissions in line with Paris Agreement commitments to preserve coral reefs.

Extreme climatic events threaten species and ecosystems across the world, with acute impacts already occurring in marine habitats[1]. If ecological functions are to be maintained in the future, then species and ecosystems will need to adjust in response to climate change[2]. It remains to be seen if ecological adjustments can keep pace with ocean warming. Coral reefs are one of the marine ecosystems most vulnerable to climate change, having faced unprecedented declines due to marine heatwaves that trigger mass coral bleaching and mortality events[3–5]. For these ecosystems to persist, assemblages of reef-building corals need increased levels of 'thermal tolerance'[6–8], here defined as the temperature threshold above which the coral assemblage experiences thermal stress. Changes in bleaching susceptibility have already occurred on some coral reefs[9–12], yet the rate at which coral thermal tolerance has and could continue to increase naturally remains unknown. It is also unclear to which extent this may occur at the

individual, species population, assemblage, or ecosystem levels. Quantifying these rates will have large ramifications for predictions of future bleaching trajectories.

Patterns in coral bleaching show that the thermal tolerance of coral assemblages can change over time. When faced with recurrent marine heatwaves and episodes of mass bleaching, individual reefs have shown higher bleaching resistance in later events[10,12–14]. Such increases in thermal tolerance can come about naturally through various mechanisms, including community composition turnover, genetic adaptation, and acclimatisation. Turnover in coral community composition away from bleaching susceptible and toward stress-tolerant coral taxa can result in increased assemblage-wide tolerance. This is suggested as one mechanism that could have decreased bleaching severity on the Great Barrier Reef in 2017 after mass bleaching and mortality the previous year, particularly due to loss of

[1]School of Natural and Environmental Sciences, Newcastle University, Newcastle upon Tyne, UK. [2]Institute of Resources, Environment and Sustainability, and Department of Geography, University of British Columbia, Vancouver, BC, Canada. [3]Marine Spatial Ecology Lab, School of Biological Sciences, The University of Queensland, St Lucia, QLD, Australia. [4]Palau International Coral Reef Center, Koror, Palau. [5]Department of Geography and Environmental Sciences, Northumbria University, Newcastle upon Tyne, UK. ✉e-mail: liamlachs@gmail.com

heat-sensitive *Acropora* and *Pocillopora*[15]. Genetic adaptation can improve species heat stress resistance over multiple generations through natural selection, increasing the frequency of genes that provide higher heat resistance and improve overall fitness[16]. Acclimatisation within an individual organism's lifetime due to environmental memory can result in higher heat resistance at the individual level[17]. Furthermore, these processes (i.e., community composition turnover, genetic adaptation, and acclimatisation) can also occur in coral's symbiotic microalgal or microbial communities[18,19]. Playing out over multiple spatial, temporal and taxonomic resolutions, these various mechanisms can all contribute to improving the thermal tolerance of coral assemblages, often precluding the assignment of causality to field observations of increased tolerance[9,10,12–15].

Adaptive mechanisms are currently only coarsely represented in future projections of mass coral bleaching[8,20], if at all[21,22]. Future bleaching scenarios are commonly derived from projections of accumulated heat stress measured as degree heating weeks (DHW)[21,22], which reflects both the duration and intensity of thermal stress[23,24]. Based on observations, mass coral bleaching is expected beyond DHWs of 4 °C-weeks, and significant mortality is expected beyond DHWs of 8 °C-weeks[23]. Although some future projection studies have included adaptation scenarios[8,20], they typically assume fixed increases in thermal tolerance across the time course of future projections. These assumptions may be biologically unrealistic, given that change in coral community composition, genetic adaptation, and acclimatisation are likely to occur non-linearly.

Here we show that the thermal tolerance of coral communities has likely increased since the late 1980s for a remote Pacific coral reef system. The rate at which thermal tolerance can increase naturally has remained largely unknown, despite its critical role in determining to what extent coral assemblages can keep pace with ocean warming. Our simulation modelling approach can address this knowledge gap for well-studied locations that have sufficient historic mass bleaching survey data. We quantify the extent that thermal tolerance has increased naturally over the past three decades in Palau, a well-studied and isolated coral reef system with extensive bleaching and ecological survey data. This is achieved by simulating 13 different historic rates of change in thermal tolerance (from 0.0 to 0.3 °C/decade, Supplementary Fig. 1) using daily 5 km satellite sea surface temperature data[25], and computing time series of bleaching heat stress (DHW) under each simulation. We then identify the most-likely historic rate of thermal tolerance increase by comparing the parsimony of DHW-derived bleaching predictions validated against historic bleaching observations[26,27]. We also assess whether high solar insolation during marine heatwaves could explain our results, given this can further exacerbate bleaching and mortality responses[28]. We contextualise how these results may alter future projections of mass coral bleaching under four contrasting global emissions scenarios, utilising statistically downscaled data from 17 global circulation models. Specifically, we explore how predictions of future bleaching trajectories change depending on the rate of thermal tolerance enhancement, with a focus on the most-likely historic rate.

## Results

### Historic mass bleaching and marine heatwaves
Since 1985, mass coral bleaching conditions have been reached across Palau in three major events. The first of these was a part of the 1998 global mass coral bleaching event which led to catastrophic mortality, with DHWs reaching a maximum of 7.1 °C-weeks (5.6 °C-weeks average across reef cells, Fig. 1a, c). Mass bleaching occurred again in 2010 corresponding to marine heatwave conditions with DHWs reaching 9.0 °C-weeks (6.4 °C-weeks average across reef cells, Fig. 1a, c). Yet, despite DHWs reaching 7.8 °C-weeks during the most recent heatwave event in 2017 (6.5 °C-weeks average across Palauan reef cells, Fig. 1a), there was little to no bleaching documented (Fig. 1b, c). Moreover, this

difference in bleaching response was unconnected to light intensity, which was similar among years. Satellite-based photosynthetically available radiation (PAR) was strongly overlapping for the August to September heatwave period among the 1998, 2010, and 2017 events, at $47 \pm 3$, $44 \pm 2$, and $44 \pm 2$ mol m$^{-2}$ day$^{-1}$ (average ± SD across all Palauan reef pixels), respectively, with significantly higher PAR in 1998, approximately 3 mol m$^{-2}$ day$^{-1}$ higher than the other two years (Supplementary Fig. 2, LMM Tukey test, $P < 0.001$).

### Simulations of coral thermal tolerance
We created 13 different thermal tolerance simulations such that tolerance increases linearly from 1988 at different rates (Supplementary Fig. 1, 0–0.3 °C/decade at intervals of 0.025 °C/decade). Despite the same measured SST history, the DHWs profiles corresponding to each simulation were strikingly different, particularly in the later years after there had been more time for thermal tolerance to shift from the baseline temperature stress threshold level. Giving the 2010 marine heatwave as an example (Fig. 2, showing DHW based on average SST of all Palauan reefs), the maximum DHW reached without any thermal tolerance enhancement was 6.6 °C-weeks (Fig. 2, yellow line). Whereas under a simulated thermal tolerance enhancement of 0.1 or 0.2 °C/decade, the maximum DHW reached was 1.7 and 0.3 °C-weeks, respectively. Under the most rapid rate of thermal tolerance enhancement simulated (0.3 °C/decade, blue line) there was no accumulation of heat stress (maximum DHW = 0) as measured SSTs never breached the theoretical temperature stress threshold (Fig. 2, dashed blue line).

### Historical increase in thermal tolerance
Based on data from all years in which bleaching surveys were conducted, DHW had a strong effect on the probability of coral bleaching for all thermal tolerance simulations (Fig. 3). Spatial correlation in model residuals would breach the 'independent observations' assumption of simple logistic regressions, but were accounted for in our study by explicitly estimating spatial variations in model uncertainty using a Bayesian approach. This corrected for under-prediction of bleaching observations in the north of Palau compared with the south (Supplementary Fig. 3). In essence, non-independence of nearby observations are accounted for by quantifying spatially correlated error in bleaching predictions and using this to adjust the error term of model, usually increasing uncertainty toward a more realistic level compared to an equivalent non-spatial model.

Among all thermal tolerance enhancement simulations (0.0–0.3 °C/decade), an enhancement rate of 0.1 °C/decade emerged as the most parsimonious outcome based on the historic data (lowest DIC; Figs. 3, 4). This simulated rate of increase in thermal tolerance was associated with the highest DHW-bleaching model parsimony (Fig. 4a), the highest prediction success rate exceeding 65% (Fig. 4b), and the lowest rate of bleaching misclassification (Fig. 4c, 15% of over-predictions and 19% of under-predictions). Despite high prediction success rates for all thermal tolerance simulations between 0.05 and 0.125 °C/decade, the most-parsimonious simulation by over 10 DIC units was for a thermal tolerance increase of 0.1 °C/decade. Under more rapid thermal tolerance enhancement simulations, bleaching was heavily misclassified by under-predictions (29%) and over-predictions (40%). This was particularly driven by the underestimation of DHW in later years due to thermal tolerance thresholds reaching exceedingly high and unrealistic levels, leading, for instance, to low DHW values in 2010 despite widespread bleaching in that year (Figs. 2, 3).

### Future bleaching projections under climate change
If mass coral bleaching conditions (i.e., DHW events >8 °C-weeks) occur two or more times per decade at any given reef, it is likely this is too often to allow for reef recovery[8]. Without increased levels of

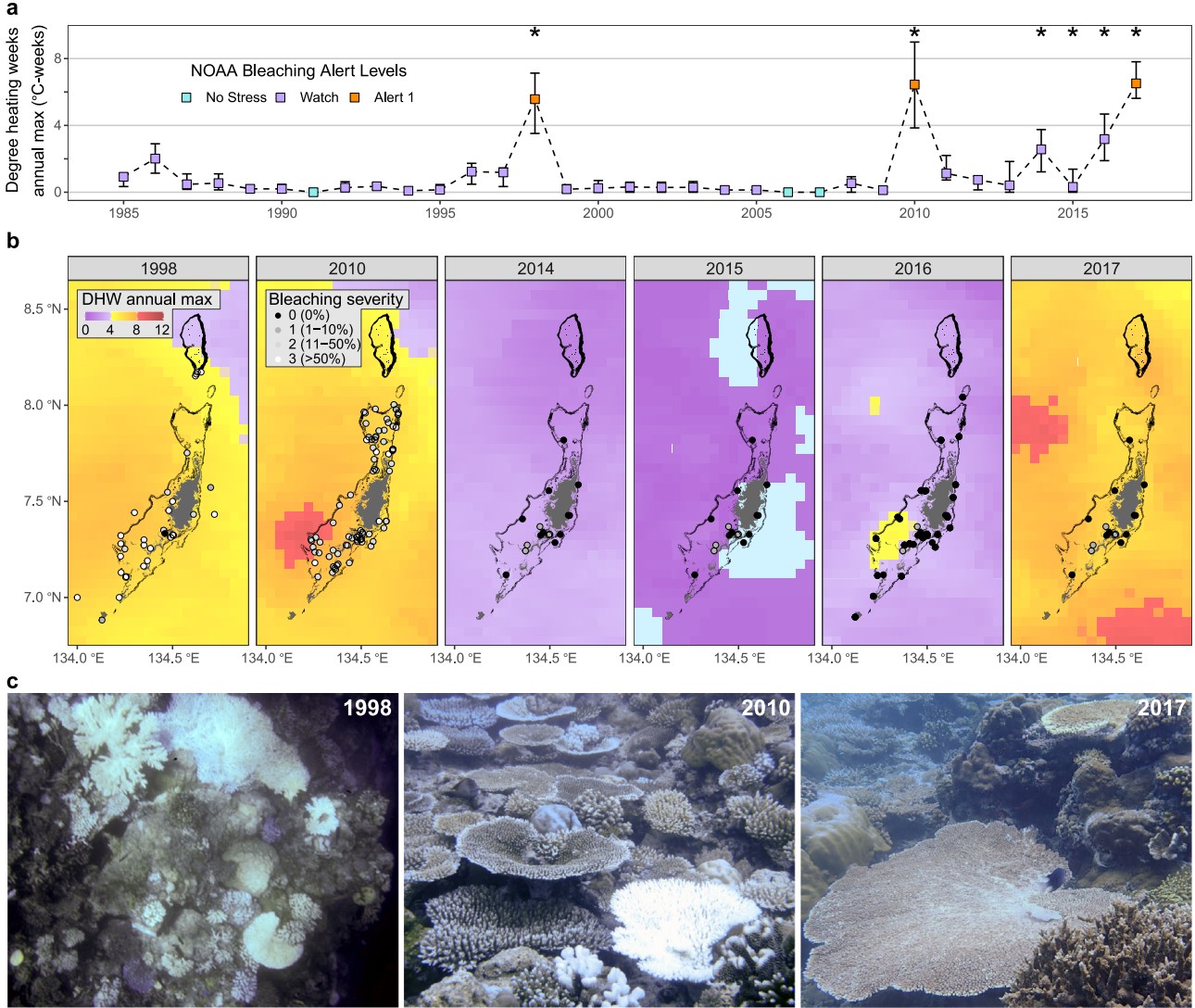

**Fig. 1 | Historic heat stress and mass coral bleaching in Palau.** Timeseries of annual maximum DHW values assuming no thermal tolerance enhancement (**a**), shown as the mean (square) and range (error bar, minimum to maximum) across all reef pixels (N = 152). Large-scale coral bleaching surveys were conducted in 1998, 2010, and 2014–2017 (years marked with *). Mass coral bleaching was recorded in 1998 and 2010, but not in 2017, despite comparable DHWs. Sporadic observations in 2001–2003 and 2006 documented no bleaching. Source data are provided as a Source Data file. Maps for each large-scale bleaching survey (**b**) show bleaching severity at survey locations (greyscale points), annual maximum DHW (raster), land area (dark grey polygons) and reef area (black polygons). Example images (**c**) show bleached coral assemblages in 1998 (photo credit: Dr. Pat Colin) and 2010 (photo credit: Dr. Robert van Woesik), and a healthy assemblage in 2017 (photo credit: Coralassist lab).

thermal tolerance (Fig. 5, yellow dashed line), such "high-frequency bleaching" (sensu[8,19]) was projected to occur by 2040 across all future emissions scenarios in our analysis. However, if the most-likely historic increase in thermal tolerance could be maintained throughout the coming century (Fig. 5, red line 0.1 °C/decade), bleaching projections were substantially reduced across all emissions scenarios. Under a scenario that reflects the temperature limits of the original Paris Agreement (SSP1-2.6), the extent of high-frequency bleaching peaks in 2050, affecting 50% of Palauan reefs, but drops by 2100 to 25% of reefs (Fig. 5a, red line). Such reductions are linked to global cooling beyond 2050 in this scenario (Fig. 5a, greyscale DHW trend).

Under all higher emissions scenarios (SSP2, SSP3, and SSP5), maintaining the most-likely historic rate of increase in thermal tolerance is sufficient to delay the onset of high-frequency bleaching across most reefs by 10–20 years compared with the counterfactual of no thermal tolerance enhancement (Fig. 5b–d, horizontal difference between yellow and red lines). However, by the end of the century even under the most-likely rate of thermal tolerance enhancement, most

reefs (Fig. 5b, SSP2) or all reefs (Fig. 5c, d, SSP3 and SSP5) become exposed to high-frequency bleaching.

For the more rapid simulated rates of thermal tolerance enhancement (0.2 and 0.3 °C/decade, Fig. 5a, b, purple and blue lines), high-frequency bleaching is absent from most reefs in both the Paris Agreement and middle-of-the-road scenarios. Under the middle-high emission scenario, only the most rapid simulated rate of thermal tolerance enhancement is sufficient to avoid bleaching impacts (Fig. 5c, 0.3 °C/decade). For the worst-case scenario, SSP5, although neither the 0.2 nor 0.3 °C/decade simulations are sufficient to prevent high-frequency bleaching altogether (Fig. 5d, purple and blue lines), they do still improve bleaching trajectories by providing multidecadal delays in the onset of high-frequency bleaching (Fig. 5d, horizontal distance between lines).

## Discussion

The substantial declines that coral reefs have experienced over the past decades are projected to continue throughout the 21st century[3,4].

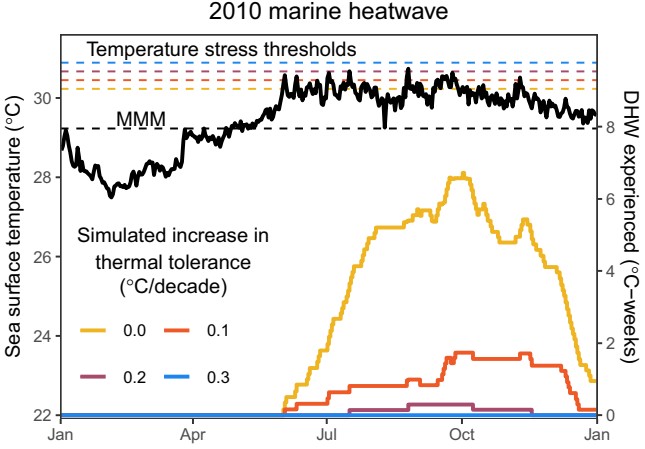

**Fig. 2 | Measured heatwave stress depends on thermal tolerance enhancement rate.** Sea surface temperature (SST) for 2010 and associated heat stress experienced (DHW) varies under different simulated rates of increase in the thermal stress threshold of corals (i.e., thermal tolerance). The black dashed line shows the maximum of monthly mean climatologies (MMM) of Palauan reefs (left y-axis) which is based on the temperature of these reefs between 1985 and 2012. The yellow dashed line is 1 °C above this MMM and represents what is currently considered to be the bleaching temperature stress threshold for corals[23,54]. The other dashed lines show the different thermal stress thresholds that would be achieved by 2010 under the simulated rates of increase in thermal tolerance of 0.1, 0.2 and 0.3 °C/decade since 1988 (left y-axis, see Supplementary Fig. 1). The black solid line is the daily sea surface temperature record for 2010 (left y-axis). If sea surface temperature is above a dotted line, that thermal stress threshold has been surpassed leading to a risk of mass coral bleaching. This risk is quantified by measuring accumulated heat stress as degree heating weeks (DHW, right y-axis), which is calculated daily from the SST that has been experienced and the simulated temperature stress threshold for that year. Source data are provided as a Source Data file.

For reefs to persist, the thermal tolerance of coral assemblages will need to increase from the levels present today, yet the natural rate at which coral thermal tolerance has already and may continue to shift has remained unknown. Here we present a simulation modelling approach that can address this knowledge gap for well-studied locations that have sufficient historic mass bleaching survey data. We show that corals in Palau did not suffer mass bleaching during the 2017 marine heatwave despite levels of heat stress and light intensity that were broadly equivalent to the conditions that led to mass bleaching in 1998 and 2010 at the same reefs. This suggests that the divergent bleaching responses among events were more likely caused by biological mechanisms rather than environmental differences. Our simulations show that a thermal tolerance increase of 0.1 °C/decade is an emergent property of Palauan coral assemblages over the past three decades. We contextualise this result under climate change by testing how further increases in thermal tolerance at this rate alter mass coral bleaching trajectories. If the historic increases in thermal tolerance can continue through the 21st century, then high-frequency bleaching could subside after 2050 under the most optimistic Paris Agreement scenario (SSP1-2.6, where global warming is limited to below 2 °C (SSP1-2.6) and may be limited substantially even under a middle-of-the-road emissions scenario (SSP2-4.5). However, high-frequency bleaching cannot be mitigated but only delayed under higher emissions scenarios (SSP3-7.0 and SSP5-8.5), where, without changes in thermal tolerance, DHWs exceed the 8 °C-weeks mass bleaching threshold annually and reach over 40 °C-weeks in some years (see[29]). Together, our study (1) adds to the documented records of increased coral thermal tolerance after bleaching events, (2) identifies the most-likely rate of thermal tolerance increase for Palau to date, and (3) shows that although shifts in thermal tolerance can help coral reefs escape the

impacts of extreme ocean heat events, this will only be possible if ambitious global commitments on carbon emissions reductions are realised.

Different bleaching susceptibility between marine heatwaves can be due to environmental and biological factors, as noted for other global regions. For example, cloudy weather may have lowered the susceptibility of corals to heat stress in the Society Islands[28], and historic exposure to a selective event in Singapore and Malaysia may have led to higher heat stress resistance in some key reef taxa due to adaptation[12]. From the environmental perspective, accumulated heat stress and light intensity were both broadly similar in Palau between the three marine heatwaves, with higher levels of DHW in 2017 and 2010 compared with 1998, suggesting that environmental differences are unlikely to explain the less severe bleaching responses during the 2017 heatwave, although other environmental variables not measured here could have also affected coral bleaching responses, for instance, water quality[30], fine-scale temperature variability[31], and thermal priming before the onset of marine heatwaves[17,32]. As such, biological processes including species composition turnover[15], genetic adaptation[33], and acclimatisation[17], in both corals and symbionts, may be more likely candidates for driving the observed increases in thermal tolerance. However, each of these mechanisms can come with potential trade-offs affecting coral populations[34,35] and reef ecosystems[36,37]. More detailed historical data on species composition, taxa-specific bleaching sensitivities, genetic bottlenecks, and prevalence/change of different symbiont taxa would be needed to quantify the relative contribution of biological drivers to the observed patterns in thermal tolerance. However, some scientific evidence is available for Palau to begin to unpack the more-likely or less-likely drivers of emergent trends.

The 1998 marine heatwave caused severe bleaching and mortality in *Acropora*, *Pocillopora* and Agariciidae, but lesser and more variable impacts in Poritidae and Favidae[38]. The *Acropora* and *Pocillopora* populations have since recovered, with coral assemblages remaining relatively stable since 2010 (except for on eastern reefs impacted by storm Bopha in 2012)[39]. On Australia's Great Barrier Reef, the bleaching susceptibility of coral assemblages decreased in the years post-2016, likely due to significant losses of heat sensitive taxa like *Acropora* and *Pocillopora*[15]. However, in Palau these taxa have recovered since 1998 and 2010[39]. Such a lack of profound shifts in coral community composition suggests that the emergence of higher thermal tolerance by 2017 may be a result of other biological mechanisms, although further genus-level or higher-taxonomic-level analyses would be required to confirm this.

Evidence suggests that thermal tolerance is widespread for *Acropora* in Palau[22,40], with considerable intrapopulation variability of this trait even at single reefs[22], and that thermal tolerance has a heritable genetic basis[33]. The mass mortality of *Acropora* and *Pocillopora* in 1998 may have acted as a selective sweep, thereby increasing the frequency of thermal tolerance genes in the remaining population. As such, genetic adaptation in certain species may have influenced increases in thermal tolerance of the wider coral assemblage. Acclimatisation of coral colonies over the past decade may have also contributed to these trends, as low levels of heat stress have occurred annually since 2010, even during La Niña years. Such conditions are in line with those suggested to shift trait values through environmental memory[17]. Moreover, corals in the inner lagoons of Palau host higher proportions of *Durisdinium* spp. symbionts which are known to confer increased levels of thermal tolerance to coral hosts[41]. As such, there is also the possibility that biological mechanisms in the symbiont community could have contributed to the apparent emergence of increased thermal tolerance in Palau.

Even remote protected reefs with few local stressors have been devastated by mass bleaching[42], leading to a growing need for climate-aware management strategies. Approaches are now being considered to boost natural adaptation rates for certain coral species, such as

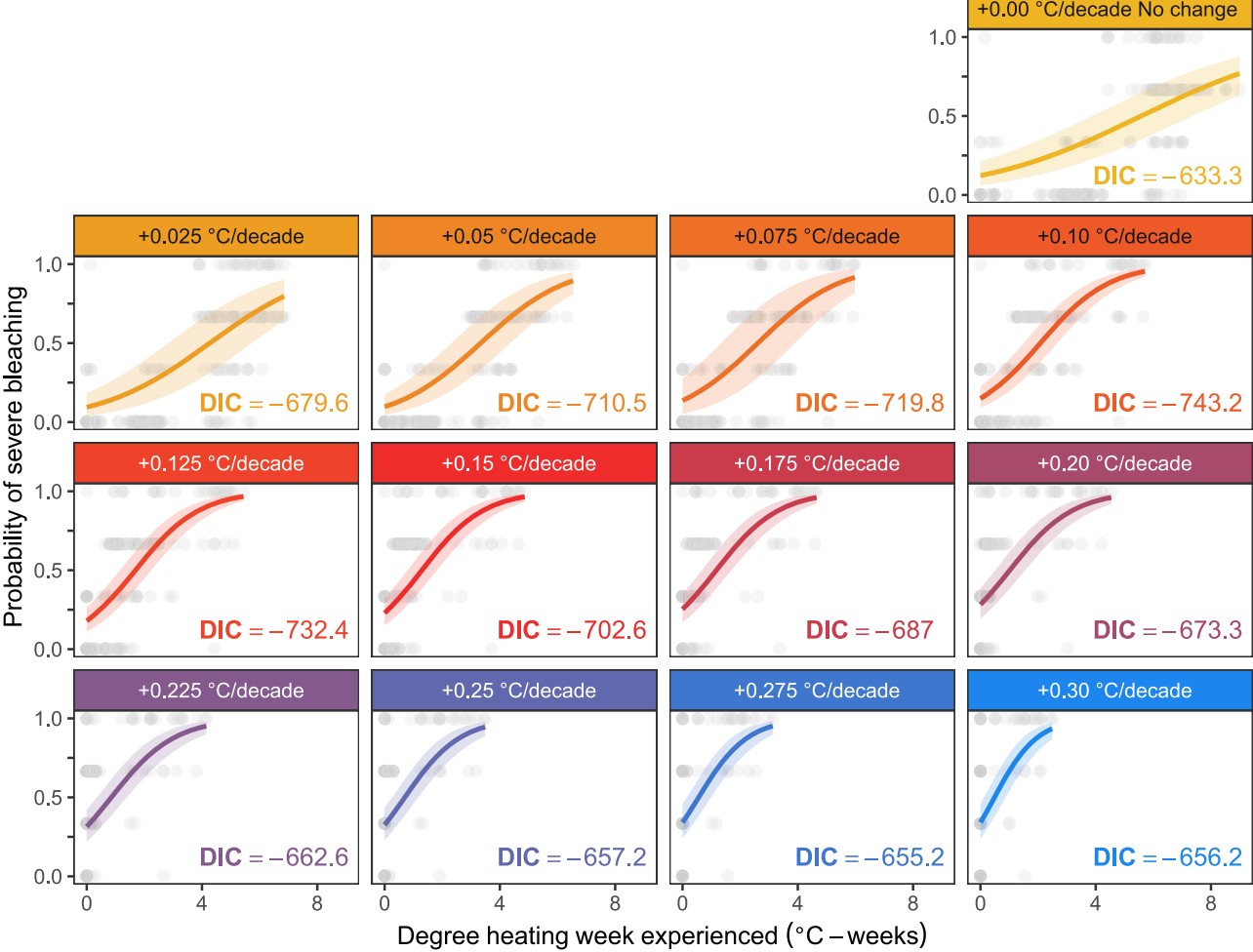

**Fig. 3 | Bleaching predictions by simulated thermal tolerance enhancement.**
Bleaching predictions based on DHW without any thermal tolerance enhancement (upper row), and with the 12 different enhancement rates from 0.025 to 0.3 °C/ decade (lower three rows), showing individual bleaching observations ($N = 237$ records) each with a unique time and location (points), model predictions (lines and uncertainty−mean and 95% credible intervals), and model parsimony, evaluated using the deviance information criterion (DIC). Notably, the most parsimonious predictions with lowest DIC were achieved for a simulated thermal tolerance enhancement rate of 0.1 °C/decade. Source data are provided as a Source Data file.

assisted evolution which aims to boost population resilience by propagating more heat tolerant corals through selective breeding or assisted geneflow[43,44]. Furthermore, protecting reefs with higher levels of coral bleaching resistance to promote natural reseeding with tolerant larvae could also benefit coral adaptation[45]. Our study has revealed an emergent increase in thermal tolerance in the corals of Palau historically, that has the potential to improve future bleaching trajectories. Additional impacts may be mitigated if natural mechanisms or management actions, such as assisted evolution, can further enhance the thermal tolerance rate of change beyond levels likely realised so far (e.g., moving from 0.1 to 0.2 °C/decade). Understanding the extent to which thermal tolerance can increase naturally on reefs and identifying the key driving mechanisms could facilitate these management approaches in defining realistic goals about the potential thermal tolerance enhancement they may be able to achieve.

Throughout the coming century, corals will need to survive progressively more intense and frequent temperature events. However, the rate at which coral thermal tolerance can increase naturally in response to climate change has been difficult to ascertain until now. Here, we reveal an emergent, assemblage-level increase in thermal tolerance that occurred over three decades. The upper limits to such rising tolerance will be set, in part, by the underlying biological mechanisms of adaptation and acclimatisation, which remain

uncertain. Thus, while we know that tolerance is increasing on decadal time scales, it remains a priority to study the diversity of potential mechanisms driving these trends. While our study demonstrates an innate ecological resilience to climate change, this is insufficient to mitigate severe impacts under middle-to-high emissions scenarios, highlighting the continued need to reduce carbon emissions and to fulfil Paris Agreement commitments.

## Methods
The Republic of Palau is located in the western Pacific Ocean and is made up of a main basalt island and southern limestone archipelago, both surrounded by a barrier reef system which extends for 120 km in the north-south direction and up to 60 km in the east-west direction. To test whether coral thermal tolerance enhancement has already occurred across Palau over the last three decades, we compiled a historic dataset of mass coral bleaching observations and 36 years of daily satellite-based sea surface temperature data. We applied spatial Bayesian statistical modelling techniques to test the susceptibility of corals to accumulated heat stress (DHW). We then contextualise the different simulated rates of thermal tolerance enhancement under climate change, testing their influence on future bleaching trajectories based on SST data from an ensemble of Global Circulation Models (GCMs). The research presented here adhered to the ethical standards

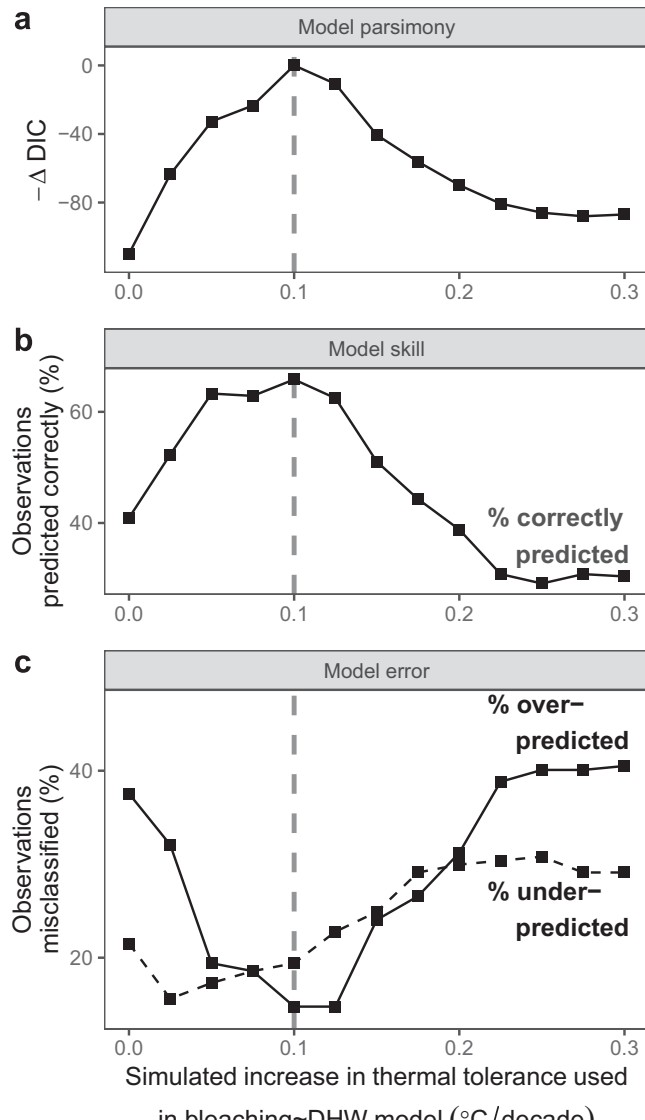

**a** Model parsimony

**b** Model skill

% correctly predicted

**c** Model error

% over−predicted

% under−predicted

Simulated increase in thermal tolerance used
in bleaching~DHW model (°C/decade)

**Fig. 4 | Model performance by simulated rate of thermal tolerance enhancement.** Comparison of the 13 bleaching prediction models (points), each with different rates of thermal tolerance enhancement and resulting bleaching heat stress (annual maximum DHW–degree heating weeks), shows: (**a**) model parsimony, as the inverse difference in deviance information criterion (DIC) between each model and the most parsimonious model; (**b**) model prediction skill; and (**c**) model prediction error, broken down into over-predictions (solid line) and under-predictions (dashed black line). Correct predictions, over-predictions and under-predictions are assigned by comparing predicted and observed bleaching severity scores. To achieve this, each predicted severity value (fitted values ranging from 0 to 1) is transformed to the closest severity score (none, mild, moderate, or severe, corresponding to values 0, 0.333, 0.667, or 1), and then compared against the corresponding true bleaching severity score of the initial observation. The most-likely historic rate of thermal tolerance enhancement from the most parsimonious model is highlighted in all panels (vertical dashed grey line). Source data are provided as a Source Data file.

consistent with the corresponding author's institutional and internal review board policies and received their approval.

### Historic environmental data
Heat stress on Palauan coral reefs was calculated from CoralTemp version 3.1, a 0.05° × 0.05° latitude-longitude resolution satellite-based Sea Surface Temperature (SST) dataset (1985 to 2020) available from the National Oceanic and Atmospheric Administration's Coral Reef

Watch (NOAA CRW)[25] and downloaded using FileZilla version 3.48.0. NOAA CRW measure accumulated heat stress with the Degree Heating Weeks metric (DHW) based on CoralTemp v3.1 (1985 to 2020). Coral reefs exposed to higher levels of DHW have a greater risk of mass bleaching and mortality[46]. As such, DHW is used by NOAA CRW to provide real-time bleaching risk forecasts, whereby DHW of 4–8 °C-weeks corresponds to the expectation of significant bleaching, and DHW > 8 °C-weeks corresponds to the expectation of significant bleaching and mortality. Here we follow the NOAA CRW methodology[23], but in brief, DHW on a given day ($DHW_i$) is computed as the sum of the last 12 weeks (84 days) of daily temperature anomalies ($HotSpots_i$) relative to a standard coral stress threshold baseline (*MMM*, maximum of monthly means) which is held constant through time. Only Hotspots > 1 °C are accumulated and are divided by 7 to make DHW a weekly metric.

$$HotSpot_i = SST_i - MMM, \; HotSpot_i \geq 0 \quad (1)$$

$$DHW_i = \sum_{n=i-83}^{i} \left( \frac{HotSpot_n}{7} \right), for \; HotSpot_n \geq 1 \quad (2)$$

Photosynthetically available radiation (PAR) as a proxy of light intensity was calculated for the three main marine heatwave years based on the monthly NASA datasets: 9 km Sea-viewing Wide Field-of-view Sensor (SeaWiFS) for 1998[47]; and 4 km Moderate-resolution Imaging Spectroradiometer (MODIS) Aqua for 2010 and 2017[48]. MODIS data were remapped to the 9 km SeaWiFS grid using bilinear interpolation to remove any effect of resolution on interannual comparisons. For the warmest period of the year when bleaching has been recorded (August-September), interannual differences in PAR were tested across all reef pixels using a linear mixed effect model with a random intercept for latitude and longitude of the reef pixel and linked to a post hoc Tukey test for pairwise interannual comparisons.

### Coral bleaching data
Coral bleaching is caused by a breakdown in the symbiosis between coral hosts and their photosynthetic algal symbionts and often leads to coral mortality[49]. Underwater coral bleaching survey observations used in this study were collected between 1998 and 2017, with spatially extensive surveys conducted during the two mass bleaching events (1998 and 2010) and four low bleaching years (2014–2017), and sporadic observations from other years (*N* = 237 records). This dataset is publicly available[26,27] and provides the timing and coordinates of each survey record. To account for multiple survey methods in the data collection (e.g., photo transects, point intercept transects, video transects), bleaching observations in the database (reported as percentage of corals bleached) are summarised as severity scores ranging from 0 to 3: 0 = no bleaching (0%), 1 = mild bleaching (1–10%), 2 = moderate bleaching (11–50%), and 3 = severe bleaching (>50%)[27].

### Bleaching prediction models
To test the effect of DHW on mass coral bleaching in Palau, we combined CoralTemp with historic coral bleaching survey observations. For each bleaching record the corresponding value of heat stress for the date of that observation was assigned from the encapsulating 5 km grid cell. For records that only reported a survey month and year, the DHW from the 15th day of the month was used. The effect of DHW on bleaching was fit using a spatial beta GLM via Integrated Nested Laplace Approximation (INLA) in R-INLA. This Bayesian statistical approach provides an intuitive spatially explicit estimation of model uncertainty, accounting for spatially correlated error[50,51]. Spatial dependencies in bleaching observations are delt with by implementing the Matérn correlation across a Gaussian Markov random field (GMRF)[52]. Essentially, this is a map of spatially

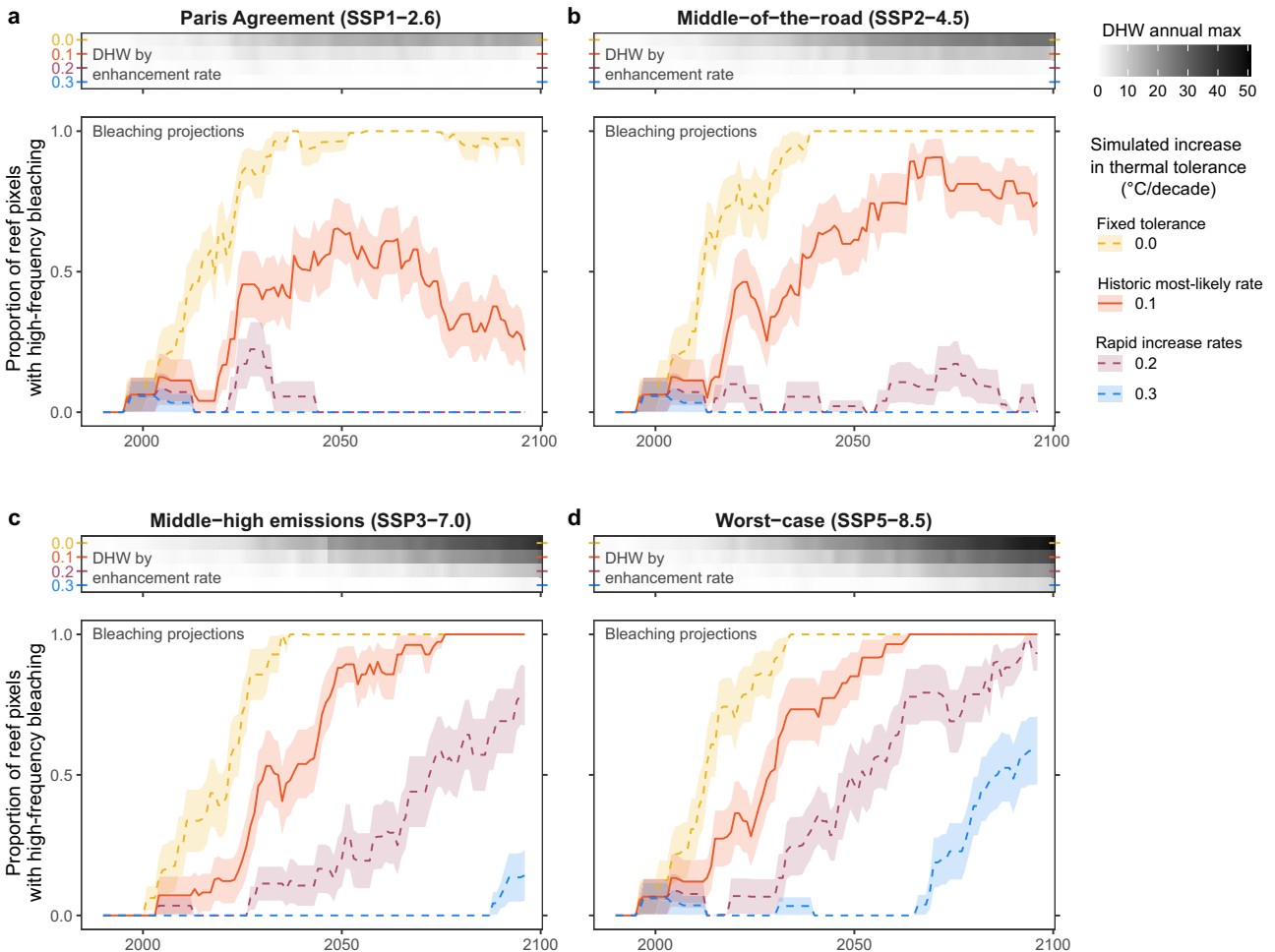

**Fig. 5 | Projected future bleaching scenarios.** Future projections of accumulated heat stress (greyscale upper panels; DHW—degree heating weeks) and mass coral bleaching conditions (lower panels) are shown for four shared socioeconomic pathways (SSP) with Paris Agreement (**a**, SSP1–2.6), middle-of-the-road (**b**, SSP2–4.5), middle-high (**c**, SSP3–7.0), and worst-case (**d**, SSP5–8.5) emissions scenarios. Ensemble projections are based on downscaled sea surface temperatures from 17 Global Circulation Models (GCMs). High-frequency bleaching is considered to occur when at least two bleaching events (i.e., DHW events >8 °C-weeks) are projected to occur in any given decade. Projections of DHW (upper panels, greyscale shading) and bleaching (lower panels, lines) are shown for four simulated increases in thermal tolerance: 0.0, 0.1, 0.2, and 0.3 °C/decade (coloured lines). A fixed thermal tolerance (yellow dashed line) is the null hypothesis, the most-likely historic rate of change in thermal tolerance according to the most parsimonious model is 0.1 °C/decade (bold line), and more rapid shifts beyond the most-likely historic rate are also shown (purple and blue dashed lines). DHW is displayed as an ensemble mean across all GCMs, while bleaching trajectories are shown as the ensemble mean ± SE. Source data are provided as a Source Data file.

correlated uncertainty that shows where bleaching is under- or over-predicted for a given heat stress dosage at the scale of individual reef sites, not the grid resolution of the heat stress fixed effect (5 km), whilst accounting for non-independence of bleaching observations that happen to occur in the same grid cells. The parameters ($\Omega$) that determine the Matérn correlation are the range ($r$ – range at which spatial correlation diminishes) and error ($\sigma$). To fit the beta distribution, bleaching severity scores were converted to proportions (by dividing by 3) and transformed to remove extremes of 0 and 1, which in practice is $(y \times (n-1) + 0.5)/n$, where n is the sample size[53]. This preserves differences among bleaching categories that would be lost using other transformations (e.g., binary transformation for a binomial GLM[54]).

Spatial variations in the uncertainty of bleaching-DHW responses were estimated across a high-resolution Delaunay triangulation mesh of the study area with a maximum triangle edge length of 2 km and a low-resolution convex hull (convex = −0.1) around the study sites to avoid boundary effects (6,210 nodes). The probability of coral bleaching for a given observation ($CB_i$) and location ($i$) was assumed to follow a beta distribution (fitted values

of $\pi_i$, and precision parameter $\theta$) using the logit-link function. Bleaching was modelled as a function of heat stress ($DHW_i$) whilst accounting for additional underlying spatial correlation among bleaching observations (random effect: $u_i$),

$$CB_i \sim \text{beta}(\pi_i, \theta) \tag{3}$$

$$\text{Expected}(CB_i) = \pi_i \tag{4}$$

$$\text{Variance}(CB_i) = \pi_i \times \frac{1 - \pi_i}{1 + \theta} \tag{5}$$

$$\text{logit}(\pi_i) = \beta_0 + \beta_1 \times DHW_i + u_i + \varepsilon_i \tag{6}$$

$$u_i \sim \text{GMRF}(0, \Omega) \tag{7}$$

$$\varepsilon_i \sim \text{N}(0, \sigma^2) \tag{8}$$

where $\beta_0$ is the intercept, $\beta_1$ is the slope for DHW, $u_i$ represents the smoothed spatial effect from the GMRF mesh, elements of $\Omega$ ($r$ and $\sigma$) are estimated from the Matérn correlation, and $\varepsilon_i$ contains the independently distributed residuals. We specified weakly informative priors for $r_0$ (20 km) and $\sigma_0$ (1) based on the residual variogram and recommendations from Zuur and Ieno[51], respectively.

## Simulated increases in thermal tolerance

Here, we simulate increases in the thermal tolerance of the coral assemblage using SST data. In the DHW algorithm, the local climatological baseline (MMM + 1 °C) represents the theoretical temperature stress threshold for corals[55]. We increased the thermal stress threshold gradually to reflect slow changes that can occur in coral communities such as turnover of the coral assemblage, natural selection, genetic adaptation of the coral host or symbiont, and symbiont shuffling. Under each simulated rate of thermal tolerance increase, we recalculated the DHW metric using the corresponding altered local climatological baseline. In practice, each baseline started to increase in 1988 (middle year of the NOAA CRW baseline period), undergoing step increases at the end of each year (when temperatures are at their seasonal low) at a magnitude that reflects the specific enhancement rate. Thirteen rates of thermal tolerance increase were computed between 0.0 and 0.3 °C/decade at equal intervals (0.025 °C/decade). This range encompasses the lower estimates of the global required rate of coral thermal tolerance enhancement of 0.2–1.0 °C/decade published by Donner et al.[8]. For each enhancement rate, the effect of DHW on bleaching was fit using a spatial beta GLM using R-INLA. The most-likely thermal tolerance enhancement rate was then determined by comparing model parsimony (DIC−Bayesian Deviance Information Criterion), bleaching prediction success rates and rate of bleaching misclassification among all 13 GLMs.

## Future projections of heat stress and mass coral bleaching

To contextualise the influence that increases in thermal tolerance would have on future projected mass coral bleaching conditions, daily SST datasets were downloaded from the World Climate Research Programme (WCRP) Coupled Model Intercomparison Project (CMIP6) using GNU Bash version 5.0.16(1) for 17 GCMs (Supplementary Tables 1, 2) across four Shared Socioeconomic Pathways: SSP1-2.6, SSP2-4.5, SSP3-7.0, and SSP5-8.5. Respectively, these climate scenarios result in 2.6, 4.5, 7.0, and 8.5 W m$^{-2}$ of radiative forcing, and 1.6, 2.8, 4.4, and 5.8 °C of global warming, by the end of the 21st century[56]. The lowest number realisation (r1 if available, realisations are climate model runs with different initial conditions) was used as the representative run of each GCM[21,57]. Each GCM SST dataset was then statistically downscaled to CoralTemp resolution[8,57] using Climate Data Operators, Bash and MATLAB on a high performance computing cluster 'The Rocket', Newcastle University. GCM SST datasets from different institutions were re-gridded to a shared GCM grid resolution using bilinear interpolation (1 ° longitude, 1/3 ° latitude) and then overlaid on the CoralTemp grid using nearest neighbour interpolation. GCM SST datasets were seasonally adjusted against the observational record (CoralTemp v3.1) using the delta method for the historical 1985 to 2010 baseline period, applying a 31-day running mean over daily climatologies to remove daily-scale noise from the annual cycle. Projections of future heat stress were then computed using the standard NOAA DHW algorithm for daily data. Following van Hooidonk et al.[21,57], years were considered to have experienced mass coral bleaching conditions if the NOAA CRW Bleaching Alert Level 2 threshold (8 °C-weeks) was reached or exceeded. Coral reef cells were identified as those that intersect coral reefs in Palau[58,59] ($N = 152$). For each year, a decadal 'high-frequency bleaching' condition was assigned to a coral reef cell if two or more years from the surrounding decade (10-year window centred on the year in question) had experienced mass bleaching conditions. This was used in place of the common 'annual severe bleaching' condition

metric because mass bleaching events every five years on average are likely to be too frequent to allow for full recovery[8,15]. The proportion of coral reef grid cells experiencing decadal high-frequency bleaching was then calculated for each year in the projection for all Palauan coral reefs.

## Limitations of thermal tolerance simulation model

Considerations that our simulation study cannot account for include hard physiological limits to thermal tolerance, associated trade-offs with other fitness-related traits, and how responses may change as corals approach their upper thermal limit. While it is notoriously difficult to accurately determine present thermal limits, let alone future changes to thermal limits for natural communities[60], this area of research will likely be important to address in the future. Furthermore, mass bleaching data only allowed us to model changes in temperature stress thresholds for coral assemblage as a whole, despite thermal thresholds likely differing both among[61] and within species[22]. As such, we cannot provide probabilities as to which rate of thermal tolerance increase is the most or least likely in the future. By modelling thermal tolerance as a gradually increasing threshold, we have improved on previous work that has fixed thermal tolerance to specific values without any rate of change[20,21,62], missing key characteristics of trait evolution and community composition shifts[39,63]. However, our model also assumes that rates cannot vary through time, which does not capture the potential non-linearities in evolutionary rates and species turnover rates, particularly when faced with punctuated stress events[63]. For example, a marine heatwave could cause selective mortality of thermally sensitive species that results in an immediate increase in thermal tolerance for the remaining coral assemblage, yet the subsequent return of sensitive species during recovery could gradually erode the previous thermal tolerance gain[15].

## Reporting summary

Further information on research design is available in the Nature Portfolio Reporting Summary linked to this article.

## Data availability

All original data has been deposited on Figshare at https://doi.org/10.25405/data.ncl.19982484[64]. The land mask for maps in this study were based on the NOAA National Centre for Coastal Ocean Science Data Collection[59], and the coral reef mask used was from the United Nations global distribution of coral reefs[58]. All datasets analysed are publicly available as of the date of publication. Source data are provided with this paper.

## Code availability

All original R code (version 4.0.2) used for data analysis is available at https://doi.org/10.25405/data.ncl.19982484[64]. Contained in the repository is sufficient data and code to reproduce all analyses in the study, however, any further guidance or additional information required is available from the lead contact upon request (e.g., downloading software/packages etc.).

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

## Acknowledgements

This works was supported by a UKRI Mitacs Globalink grant to L.L., J.R.G, and S.D.D. (NE/T014547/1), the Natural Environment Research Council's ONE Planet Doctoral Training Partnership Studentship (NE/S007512/1) to L.L., and a European Research Council Horizon 2020 project CORALASSIST (725848) to J.R.G. We also thank Dr. Harmony Martell, Dr. Pedro Gonzalez-Espinosa, and Dr. Xinru Li for their thoughts on this work, and Dr. Yimnang Golbuu for supporting our research at the Palau International Coral Reef Centre.

## Author contributions

L.L., S.D.D., and J.R.G. conceived and designed the study; L.L. developed the computer code; L.L., analysed the results; L.L. wrote the first draft of the paper; L.L., S.D.D., P.J.M., A.H., J.C.B., H.K.E., and J.R.G critically revised the manuscript.

## Competing interests

The authors declare no competing interests.
