## [Peer Review File · Nature Communications]

Emergent increase in coral thermal tolerance reduces mass bleaching under climate changeREVIEWER COMMENTS

Reviewer #1 (Remarks to the Author):

Review of Emergent increase in coral thermal tolerance reduces mass bleaching under climate change

The key finding of this study is that Palauan coral reef assemblages have likely experienced a historical increase in thermal tolerance of 0.1°C per decade since 1998. This is a significant result, as it is an important step towards understanding the extent to which natural adaptation may keep pace with climate change. It is also a key parameter to inform global predictions of reef futures which is especially valuable given the scarcity of literature quantifying such values. Additionally, the methodology used in this study has the potential to be applied to other reef systems which would lead to more interesting results.

While the mechanisms that likely lead to the increase in tolerance cannot be determined with certainty from the available data, the authors present a comprehensive discussion of the main potential mechanisms and comment on which may be most likely.

Therefore I believe this research is of high significance to the fields of coral reef ecology, climate science, evolutionary ecology, ecological modelling and more widely. I recommend publication in Nature Communications if my comments can be addressed.

I have made minor comments and suggestions to improve clarity throughout the manuscript detailed by line number.

I have three more detailed comments I would want to see addressed before publication.

1) Bleaching as a percentage metric

If bleaching is measured based on the percentage of total substrate that is bleached or dead coral (as stated lines line 436) following/during a temperature stress event, then it is dependent on the pre-disturbance presence of live coral that can be killed or bleached by a disturbance.

To illustrate this in a very extreme example, prior to 1998 you could have had 80% live coral cover, and an extreme mass bleaching event could have caused total bleaching/mortality, such that you to say 80% of the reef is bleached coral. If all this coral died and did not recover fully prior to 2010 there could be e.g. 40% live coral cover prior to 2010, and again if

all of this coral bleached/died, this time you would say 40% bleaching (i.e. half the amount of bleaching as was observed in 1998, potentially leading you to assume some increase in tolerance, but actually it was just that there was half the amount of coral).

(If I have misinterpreted this part of the process then please clarify the methods text for 'Coral bleaching data'.)

In Gouezo et al 2018 it looks like the coral cover prior to disturbance is fairly similar for the three major temperature stress events, although for example the coral cover in 'outer reef east' in Gouezo et al 2018 is significantly higher prior to the 2010 event than it is prior to the 2017 heat stress event which could lead to a lesser version of the extreme example given above.

While this may not be an issue in the case of the present study if total coral cover is generally comparable prior to disturbance, I think it had the potential to be an issue and therefore needs to be clarified and discussed in the present paper as this methodology could be applied to other situations.

Generally I think a bit more detail/clarity is needed in this methods paragraph for 'Coral bleaching data'.

2) Terminology on the most likely increase in thermal tolerance

I think the wording "'historically-realised' rate of change in thermal tolerance" is too strong given you assign this from the most parsimonious model. In theory there could be other covariates and other models that were more parsimonious, but you didn't have the data to test for these. I am comfortable with the narrative that this best model and the correlation means it is the most likely rate to have occurred, but I don't think you can say this is unequivocally the 'realised rate'.

I think the terminology you use in Discussion paragraph 1 is better, i.e. "(2) identifies the most-likely rate of thermal tolerance increase for Palau to date".

3) Conclusion that a shift in species composition is unlikely the cause of increased tolerance

Line 289. "which is unlikely to be caused by shifts in species composition toward hardier coral taxa"

I think this needs some further qualification: e.g. “which we suggest shifts in species composition is unlikely to be the major mechanism, given composition of genera was fairly consistent (Gouezo 2018). However, species level analysis would be required to confirm this.”

The paper would benefit from a simple plot of coral genera composition before each of the major disturbance events, in a similar way to what you have for the light in Extended data figure 2.

Minor comments and suggestions

Methods: The methods would benefit from a brief description of the Palauan reef system (size, number of reefs, etc), ocean basin.

Discussion:

- Consider some comment on the 8DHW you discuss as definition of high frequency bleaching when occurring more than twice per decade, and the much larger >>>8DHW that are predicted under some of the SSPs?
- Consider some mention of potential trade-offs to this increased thermal tolerance?

Line 16. ‘intense and more frequent’ ? because you consider ‘high frequency bleaching’ in analyses

Line 20. ‘and/or’ shifts in community structure

Line 38. Could add a sentence something like: “It is also unclear the extent to which this has or will occur at the individual, species, population assemblage or ecosystem level”

Line 44. I thought points (1) to (4) here were a great summary of the different possible mechanisms

Line 49. Could put “...natural selection of genes that provide...”

Line 59. Suggest: Acclimation within an ‘individual’ organisms lifetime..

Line 69. Suggest ‘to what extent’ instead of ‘whether’

Line 74. ‘Simulating 13 different’ not ‘13 simulating’

Line 82. Suggest: “Specifically, we explore how ‘predictions of’ future bleaching trajectories ‘for Palauan reefs’ change depending on the thermal tolerance enhancement rate, with a focus on ‘what we identify as the most likely rate of increase in tolerance’

Line 90-92. Suggest rearranging this sentence to start with second half “Despite DHWs reaching...”

Line 93. This is the first mention of light intensity, consider mentioning in the Introduction that it is another potentially important driver of bleaching likelihood.

Line 102 (Fig1. Legend): Suggest slight rewording: Large-scale coral bleaching surveys (*) were conducted in 1998, 2010, and 2014–2017. “Mass bleaching was recorded on these surveys in 1998 and 2010, but not following 2010 despite comparable DHWs.”

Line 104. Suggest rewording to: “Sporadic observations in 2001–2003 and 2006 documented no bleaching”

Line 106. Add ‘at point survey locations’ (if this is true, see other comments on bleaching severity metric, need some more information in Methods)

Line 106. Not sure what you mean by outer reef area, could you say ‘reef boundary’ or do you mean something specific here?

Figure 2 legend: This figure is very important but also very complex for anyone who doesn’t understand well the relationships between MMM, SST and DHW. This relationship is provided in the appendix but I suggest some modification to this figure legend to improve clarity in the main results here. I would suggest something rearranged to read along the lines of (authors can improve but I think this narrative is easier to follow):

“Sea surface temperature and associated heat stress (DHW experienced) under different assumptions of rate of increase in temperature stress tolerance for 2010. The black dashed line shows the maximum monthly mean (MMM) of Palauan reefs (which is dependent on the temperature experienced by these reefs between 1985 and 2012, left Y-axis). The yellow dashed line is 1 deg celsius above this MMM, and represents what is currently considered to be the bleaching threshold (REFERENCE). The other dashed lines show the different thermal stress thresholds that would be achieved by 2010 under the simulated rates of increase in thermal tolerance of 0.1, 0.2 and 0.3 deg cel/decade since 1998 (left Y-

axis, see Extended Data Fig 1). The black solid line is the sea surface temperature (left Y-axis) during 2010. If the sea surface temperature is above the dotted lines, it means the thermal stress threshold has been surpassed and mass bleaching would be expected. The right Y-axis gives degree heating weeks (DHW) which was calculated from the simulated temperature stress threshold in 2010 and the sea surface temperature experienced.”

Line 141. Say what the uncertainty is, 95% credible/ confidence interval?

Figure 3 legend: say what each point is, i.e. what spatial/ temporal resolution? Assume annual for a survey site?

Figure 4c. Am I correct in interpreting that this does not consider the magnitude of under or over estimation, just categorical?

Figure 4b. May be worth putting in fig legend what defines ‘correctly predicted’

Line 166. Add reference for these ‘projection models’. Also maybe you mean “commonly evaluate the consequences of if mass bleaching occurs”... ? Need more clarity on what you mean by ‘projection models’ here.

Figure 5. I am confused by the x-axis label ‘middle year of decade’ as the temporal resolution seems to show you have a measure for each year?

Line 204: see major comment on terminology for this, suggest e.g “according to the most parsimonious model” or similar

Line 208. Suggest ‘experienced’ rather than ‘faced’

Line 215. Suggest: ‘Despite levels of heat stress and light intensity that were broadly equivalent to the ‘conditions that led to’ mass bleaching in 1998 and 2010 at the same reefs’

Line 224. Be clear that the mitigation is more under Paris than under 4.5. Also worth

mentioning that bleaching likelihood actually starts to decrease again under the Paris towards end of century.

Line 232. Name some of the other 'global regions' here?

Line 252. I think you mean bleaching susceptibility decreased, or bleaching tolerance increased?

Line 254. Stick with 'taxa' or genera, rather than species. Unless someone monitored at species level, shifts in species composition could have occurred within the Acropora or Pocillopora which would be another potential explanation for increased heat tolerance. I think worth noting this here, see major comment.

Line 257. I am unclear what you mean by 'individual heat tolerance'

Line 272. Add reference(s) – e.g. Gilmour et al 2013, Recovery of an isolated coral reef system following severe disturbance

Line 282. 'Understanding the extent to which thermal tolerance can increase naturally on reefs' ... would be good to also work mechanisms into this sentence, i.e. 'and identifying the key mechanisms allowing this increase'

Line 286. Suggest: more extreme and 'more frequent'. Also mention chronic increases in ocean temperature?

Line 423. For 'the' warmest

Line 435. Briefly mention some of these survey methods, e.g. phototransects, point surveys?

Line 438. Suggest: from 0 to 3: 0 = no bleaching (0%), 1 = mild (1-10%), 2 = moderate (11-50%) and 3 = severe bleaching (>50%).

Line 476. Add a reference, maybe (Glynn and D'Croze, 1990)

https://coralreefwatch.noaa.gov/product/5km/tutorial/crw10a_dhw_product.php

Line 497. Need to say what a realisation is

Extended Data Fig1. Could add vertical line at 2010 and refer to Figure 2 in results

Extended data Table. Insert 'Summary of GCMs used in table legend rather than 'GCM data' because this table doesn't contain the data.

Reviewer #2 (Remarks to the Author):

This is an important manuscript. There are however a few issues that need clarification, particularly concerning some of the methods of coral bleaching. The authors have provided detailed methods of sea-surface temperature and the simulations and projections, which are all clear, however, they gloss over the major response variable, which is coral bleaching. The highest resolution of SST appears to be 5 km, and we can assume that coral bleaching was observed at reef sites, recorded as geographical coordinates, but there is no mention of how the authors coupled the predictor variable (SST) and the response variable (bleaching). How many SST pixels were there for Palau and how many sites were there? It seems that some SST pixels will contain more than one study site (coordinate), if so what did the authors do to avoid pseudo-replication? Were the coral bleaching values averaged for each pixel? This is particularly important so that readers can understand Figure 4, the "observations" predicted correctly.

Some other concerns.

Abstract. Line two should read more intensive (not intense)

Change the 5th sentence to read. This led to less severe bleaching impacts than predicted, indicating...

High-frequency bleaching (compound adjectives are hyphenated)

Throughout the manuscript, compared to should read compared with
Write in parallel, 3rd to last sentence, “yet can be only delayed...”

Main text

A few important references are missing showing higher resistance to repeated coral bleaching, including Diane Thompson’s work.

The last sentence of the first page should read: “Such increases in thermal tolerance can come about naturally through various mechanisms, including turnover, genetic adaptation, and acclimatisation.” In that way, the authors do not need to awkwardly enumerate the following paragraph.

The second page, 3rd paragraph.

Should read: “This is achieved by simulating...”

Page 4.

This section needs clarification as it is confusing which data were actual measurements and which data were simulated. I assume the 2010 SST data were actual measurements. As it is written, it appears that the 2010 marine heatwaves were simulations.

Should read: “Under a simulated thermal tolerance enhancement...”

Figure 2. The actual data measurements and simulations need to be clarified here as well.

Figure 2, should not be called “Example mean...”, simply “Mean sea surface temperature...”

The historically realized increase in thermal tolerance section. The sentence on spatial correlation needs to be rewritten. How were they accounted for?

Figure 3 caption. Clarify. And model parsimony was evaluated using DIC

Figures 4B and C. Need to clarify how the response variable was estimated (see opening comment).

Discussion. Paragraph 3. The bleaching susceptibility should read “the bleaching susceptibility of coral assemblages decreased” because if sensitive taxa are lost the system becomes more tolerant, not more susceptible.

Paragraph 4. The authors are assuming that there were residual populations.

Paragraph 6. Delete the first sentence. Rewrite the awkward final sentence of this paragraph.

Methods. Bleaching prediction models. Delete “provides a more intuitive” (more intuitive than what? A frequentist model, yes, but that is unclear), simply write: “...statistical approach to spatially explicit...”

For the equation, remove GMRF, and replace it with N for the spatial mu component.

We would like to thank the two anonymous reviewers for your interest in our work, and for helpful and constructive comments. In this response to reviewers document, please note that all author responses are shown in **blue**. Quotes from the manuscript are shown in *italics*. Line numbers are shown for the simple view of the revised manuscript (tracked changes not showing), and revisions/additions compared to the last draft are shown as *italics*.

Reviewer #1 (Remarks to the Author):

Review of Emergent increase in coral thermal tolerance reduces mass bleaching under climate change

The key finding of this study is that Palauan coral reef assemblages have likely experienced a historical increase in thermal tolerance of 0.1°C per decade since 1998. This is a significant result, as it is an important step towards understanding the extent to which natural adaptation may keep pace with climate change. It is also a key parameter to inform global predictions of reef futures which is especially valuable given the scarcity of literature quantifying such values. Additionally, the methodology used in this study has the potential to be applied to other reef systems which would lead to more interesting results.

Response: We thank the reviewer for the supportive comments, and in fact we are currently implementing this work for other locations globally.

While the mechanisms that likely lead to the increase in tolerance cannot be determined with certainty from the available data, the authors present a comprehensive discussion of the main potential mechanisms and comment on which may be most likely.

Therefore I believe this research is of high significance to the fields of coral reef ecology, climate science, evolutionary ecology, ecological modelling and more widely. I recommend publication in Nature Communications if my comments can be addressed.

I have made minor comments and suggestions to improve clarity throughout the manuscript detailed by line number.

I have three more detailed comments I would want to see addressed before publication.

1) Bleaching as a percentage metric

If bleaching is measured based on the percentage of total substrate that is bleached or dead coral (as stated lines line 436) following/during a temperature stress event, then it is dependent on the pre-disturbance presence of live coral that can be killed or bleached by a disturbance.

To illustrate this in a very extreme example, prior to 1998 you could have had 80% live coral cover, and an extreme mass bleaching event could have caused total bleaching/mortality, such that you to say 80% of the reef is bleached coral. If all this coral died and did not recover fully prior to 2010 there could be e.g. 40% live coral cover prior to 2010, and again if all of this coral bleached/died, this time you would say 40% bleaching (i.e. half the amount of bleaching as was observed in 1998, potentially leading you to assume some increase in tolerance, but actually it was just that there was half the amount of coral).

(If I have misinterpreted this part of the process then please clarify the methods text for 'Coral bleaching data'.)

In Gouezo et al 2018 it looks like the coral cover prior to disturbance is fairly similar for the three major temperature stress events, although for example the coral cover in 'outer reef

east' in Gouezo et al 2018 is significantly higher prior to the 2010 event than it is prior to the 2017 heat stress event which could lead to a lesser version of the extreme example given above.

While this may not be an issue in the case of the present study if total coral cover is generally comparable prior to disturbance, I think it had the potential to be an issue and therefore needs to be clarified and discussed in the present paper as this methodology could be applied to other situations.

Generally I think a bit more detail/clarity is needed in this methods paragraph for 'Coral bleaching data'.

Response: We thank the reviewer for this comment which makes a good point. We apologise for the misunderstanding in the methodology about the bleaching data and have updated it accordingly. It wasn't a misinterpretation by the reviewer, but an incorrect statement in our methodology section. The bleaching data are from various sources and thus represent different specific metrics for each study. However, they broadly follow the Reefbase protocol (Urcelay & Donner 2023) by reporting bleaching as percentage of corals bleached, not percentage of reef area bleached (which would as reviewer 1 mentioned be confounded by coral cover and would never exceed the percentage coral coverage). To address this, we have updated the coral bleaching methodology section as follows:

Lines 487-489: *multiple survey methods in the data collection (e.g., photo transects, point intercept transects, video transects), bleaching observations in the database (reported as percentage of reef-area corals bleached), are summarised as ...*

2) Terminology on the most likely increase in thermal tolerance

I think the wording "'historically-realised' rate of change in thermal tolerance" is too strong given you assign this from the most parsimonious model. In theory there could be other covariates and other models that were more parsimonious, but you didn't have the data to test for these. I am comfortable with the narrative that this best model and the correlation means it is the most likely rate to have occurred, but I don't think you can say this is unequivocally the 'realised rate'.

I think the terminology you use in Discussion paragraph 1 is better, i.e. "(2) identifies the most-likely rate of thermal tolerance increase for Palau to date".

Response: We thank the reviewer for highlighting this issue. We have addressed this by removing reference to the term 'realised rate'. Instead, we use terms such as 'most-likely historic rate of increase in tolerance', and 'historic increase'. There were several places this was changed throughout the manuscript.

3) Conclusion that a shift in species composition is unlikely the cause of increased tolerance

Line 289. "which is unlikely to be caused by shifts in species composition toward hardier coral taxa". I think this needs some further qualification: e.g. "which we suggest shifts in species composition is unlikely to be the major mechanism, given composition of genera was fairly consistent (Gouezo 2018). However, species level analysis would be required to confirm this." The paper would benefit from a simple plot of coral genera composition before each of the major disturbance events, in a similar way to what you have for the light in Extended data figure 2.

Response: This is a good idea, and we partially already addressed this earlier in the discussion with the sentence: 'On the Great Barrier Reef of Australia, the bleaching

susceptibility of coral assemblages decreased in the years post-2016, likely due to significant losses of heat sensitive taxa like *Acropora* and *Pocillopora*¹⁵. However, in Palau these taxa have recovered since 1998 and 2010³⁸. Such a lack of profound shifts in coral community composition suggests that the emergence of higher thermal tolerance by 2017 may be a result of other biological mechanisms.’ To address your comment at this part of the discussion, we add a caveat to this statement (see below).

In the final paragraph of the discussion, we had reiterated the unlikely effect of community composition change on explaining tolerance shifts. However, this is not a major finding of our study and therefore we have removed this from sentence to address your comment. Notably, we do not have data on community composition so cannot add a supplementary data figure as suggested.

Lines 281-286: *... heat sensitive taxa like Acropora and Pocillopora¹⁵. However, in Palau these taxa have recovered since 1998 and 2010³⁸. Such a lack of profound shifts in coral community composition suggests that the emergence of higher thermal tolerance by 2017 may be a result of other biological mechanisms, although further genus-level or higher taxonomic level analyses would be required to confirm this.*

Lines 318-321: *Here, we reveal an emergent, assemblage-level increase in thermal tolerance that occurred over three decades, which is unlikely to be caused by shifts in species composition toward hardier coral taxa.*

Minor comments and suggestions

Methods: The methods would benefit from a brief description of the Palauan reef system (size, number of reefs, etc), ocean basin.

Response: We have added text in the first two paragraphs to briefly describe the Palau reef system (size, geology, ocean basin) and mention the number of reef grid cells present (in terms of 5km grid cells on which the future projection analysis is based).

Lines 442-445: *The Republic of Palau is located in the western Pacific Ocean and is made up of a main basalt island and southern limestone archipelago, both surrounded by a barrier reef system which extends for 120km in the north-south direction and up to 60km in the east-west direction.*

Lines 569-570: *Coral reef cells were identified as those that intersect coral reefs in Palau^{56,57} (N=152).*

Discussion:

- Consider some comment on the 8DHW you discuss as definition of high frequency bleaching when occurring more than twice per decade, and the much larger >>>8DHW that are predicted under some of the SSPs?

Response: We have made the suggested change.

Lines 247-250: *However, high-frequency bleaching cannot be mitigated but only delayed under higher emissions scenarios (SSP3-7.0 and SSP5-8.5), where without changes in thermal tolerance, DHWs events exceed the 8 °C-weeks mass bleaching threshold annually and reach over 40 °C-weeks in some years (see ²⁹).*

- Consider some mention of potential trade-offs to this increased thermal tolerance?

Response: We have introduced this idea in the discussion, referencing studies which quantify how different mechanisms could be associated with different trade-offs. For example, more heat tolerant symbiont could lead to lower coral growth rates which is a problem at the population scale (Cunning 2015) and ecosystem scale (Ortiz 2013), and that loss of branching corals may lead to a higher tolerant community but at the cost of habitat complexity (Magel 2019). Readers can follow these citations to learn more on these specific topics.

Lines 269-271: However, each of these mechanisms can come with potential trade-offs affecting coral populations³⁴ and reef ecosystems^{35,36}.

Line 16. 'intense and more frequent' ? because you consider 'high frequency bleaching' in analyses

Response: We have made the suggested change.

Lines 16: *corals will need to endure progressively more intense and frequent marine heatwaves.*

Line 20. 'and/or' shifts in community structure

Response: We think that and/or is not necessary here and takes away from the flow of the abstract. By saying just 'or' we do not imply that these mechanisms are mutually exclusive because we have not inserted a word like 'either' at earlier in the clause. Therefore, we have left this sentence in the original form.

Line 38. Could add a sentence something like: "It is also unclear the extent to which this has or will occur at the individual, species, population assemblage or ecosystem level"

Response: We have made the suggested change.

Lines 39-41: It is also unclear to which extent this may occur at the individual, species, population assemblage or ecosystem levels.

Line 44. I thought points (1) to (4) here were a great summary of the different possible mechanisms

Response: Thanks.

Line 49. Could put "...natural selection of genes that provide..."

Response: We have made the suggested change.

Lines 53: *natural selection of genes that provide.*

Line 59. Suggest: Acclimation within an 'individual' organisms lifetime..

Response: We have made the suggested change.

Lines 53-54: *Acclimatisation within an individual organism's lifetime.*

Line 69. Suggest 'to what extent' instead of 'whether'

Response: We have made the suggested change.

Lines 73: *critical role in determining to what extent coral assemblages can keep pace.*

Line 74. 'Simulating 13 different' not '13 simulating'

Response: We have made the suggested change.

Lines 78: *by simulating 13 different.*

Line 82. Suggest: “Specifically, we explore how ‘predictions of’ future bleaching trajectories ‘for Palauan reefs’ change depending on the thermal tolerance enhancement rate, with a focus on ‘what we identify as the most likely rate of increase in tolerance’

Response: We made some small changes to the suggestion.

Lines 87-89: *Specifically, we explore how predictions of future bleaching trajectories change depending on the thermal tolerance enhancement rate, with a focus on the most-likely historic rate.*

Line 90-92. Suggest rearranging this sentence to start with second half “Despite DHWs reaching...”

Response: We have made the suggested change.

Lines 96-98: *Yet despite DHWs reaching 7.8 °C-weeks during the most recent heatwave event in 2017 (6.5 °C-weeks average across Palauan reef cells, Fig. 1A), there was little to no bleaching (Fig. 1B,C).*

Line 93. This is the first mention of light intensity, consider mentioning in the Introduction that it is another potentially important driver of bleaching likelihood.

Response: We have made the suggested change.

Lines 83-85: *We also assess whether high solar insolation during marine heatwaves could explain our results, given this can further exacerbate bleaching and mortality responses²⁸.*

Line 102 (Fig1. Legend): Suggest slight rewording: Large-scale coral bleaching surveys (*) were conducted in 1998, 2010, and 2014–2017. “Mass bleaching was recorded on these surveys in 1998 and 2010, but not following 2010 despite comparable DHWs.”

Response: We have made the suggested change.

Lines 109-110: *Mass coral bleaching was recorded in 1998 and 2010, but not in 2017, despite comparable DHWs.*

Line 104. Suggest rewording to: “Sporadic observations in 2001–2003 and 2006 documented no bleaching”

Response: We have made the suggested change.

Lines 110-111: *Sporadic observations in 2001–2003 and 2006 documented no bleaching.*

Line 106. Add ‘at point survey locations’ (if this is true, see other comments on bleaching severity metric, need some more information in Methods)

Response: We have made the suggested change.

Lines 111-112: *Maps for each large-scale bleaching survey show bleaching severity at survey locations (greyscale points),*

Line 106. Not sure what you mean by outer reef area, could you say ‘reef boundary’ or do you mean something specific here?

Response: Yes, you are right here. We meant the reef area.

Lines 113: *and reef area (black polygons).*

Figure 2 legend: This figure is very important but also very complex for anyone who doesn't understand well the relationships between MMM, SST and DHW. This relationship is provided in the appendix but I suggest some modification to this figure legend to improve clarity in the main results here. I would suggest something rearranged to read along the lines of (authors can improve but I think this narrative is easier to follow):

“Sea surface temperature and associated heat stress (DHW experienced) under different assumptions of rate of increase in temperature stress tolerance for 2010. The black dashed line shows the maximum monthly mean (MMM) of Palauan reefs (which is dependent on the temperature experienced by these reefs ?between 1985 and 2012, left Y-axis). The yellow dashed line is 1 deg celsius above this MMM, and represents what is currently considered to be the bleaching threshold (REFERENCE). The other dashed lines show the different thermal stress thresholds that would be achieved by 2010 under the simulated rates of increase in thermal tolerance of 0.1, 0.2 and 0.3 deg cel/decade since 1998 (left Y-axis, see Extended Data Fig 1). The black solid line is the sea surface temperature (left Y-axis) during 2010. If the sea surface temperature is above the dotted lines, it means the thermal stress threshold has been surpassed and mass bleaching would be expected. The right Y-axis gives degree heating weeks (DHW) which was calculated from the simulated temperature stress threshold in 2010 and the sea surface temperature experienced.”

Response: We have revised the figure caption with some edits based on your proposed outline.

Lines 129-142: Figure 2. Sea surface temperature (SST) for 2010 and associated heat stress experienced (DHW) under different simulated rates of increase in the thermal stress threshold of corals (thermal tolerance). The black dashed line shows the maximum of monthly mean climatologies (MMM) of Palauan reefs (left y-axis) which is based on the temperature of these reefs between 1985 and 2012. The yellow dashed line is 1 °C above this MMM and represents what is currently considered to be the bleaching temperature stress threshold for corals^{22,27}. The other dashed lines show the different thermal stress thresholds that would be achieved by 2010 under the simulated rates of increase in thermal tolerance of 0.1, 0.2 and 0.3 °C/decade since 1988 (left y-axis, see Extended Data Fig 1). The black solid line is the daily sea surface temperature record for 2010 (left Y-axis). If sea surface temperature is above a dotted line, that thermal stress threshold has been surpassed leading to a risk of mass coral bleaching. This risk is quantified by measuring accumulated heat stress as degree heating weeks (DHW, right y-axis), which is calculated daily from the SST that has been experienced and the simulated temperature stress thresholds for that year.

Line 141. Say what the uncertainty is, 95% credible/ confidence interval?

Response: We have made the suggested change.

Lines 155: model predictions (lines and uncertainty – 95% credible intervals).

Figure 3 legend: say what each point is, i.e. what spatial/ temporal resolution? Assume annual for a survey site?

Response: Each point is an individual observation with specific coordinates and timestamp derived from the global bleaching database used here.

Lines 154-155: showing individual bleaching observations each with a unique time and location (points).

Figure 4c. Am I correct in interpreting that this does not consider the magnitude of under or over estimation, just categorical?

Response: Yes, in this case it can only be underpredicted, overpredicted, or correctly predicted. The bleaching severity scores are none, mild, moderate and severe, and correspond to values of 0, 0.333, 0.667, and 1, respectively. Therefore, for a record in the database showing mild bleaching (0.333), then a correct prediction is only assigned if the probability of severe bleaching (the model fitted value) is closer to that value (0.333) than any other (i.e., between 0.167 and 0.5). In this sense, the probability bands for each severity score level are as follows:

No bleaching (severe bleaching probability: 0 – 0.167)

Mild bleaching (severe bleaching probability: 0.167 – 0.5)

Moderate bleaching (severe bleaching probability: 0.5 – 0.833)

Severe bleaching (severe bleaching probability: 0.833 – 1)

Lines 178-183: Correct predictions, over-predictions and under-predictions are assigned by comparing predicted and observed bleaching severity scores. To achieve this, each predicted severity value (fitted values ranging from 0 to 1) is transformed to the closest severity score (none, mild, moderate, or severe, corresponding to values 0, 0.333, 0.667, or 1), and then compared against the corresponding true bleaching severity score of the initial observation.

Figure 4b. May be worth putting in fig legend what defines ‘correctly predicted’

Response: We have defined this in the figure caption. See comment and additional text above.

Line 166. Add reference for these ‘projection models’. Also maybe you mean “commonly evaluate the consequences of if mass bleaching occurs”... ? Need more clarity on what you mean by ‘projection models’ here.

Response: The purpose of this sentence was to define ‘high frequency bleaching’ in the main text. Therefore, we have restructured the sentence to reflect this.

Lines 187-191: If mass coral bleaching conditions (i.e., DHW events > 8 °C-weeks) occur two or more times per decade at any given reef, it is likely this is too often to allow for reef recovery⁸. Without increased levels of thermal tolerance (Fig. 5, yellow dashed line), such ‘high frequency bleaching’ (sensu^{8,18}) was projected to occur by 2040 across all future emissions scenarios in our analysis.

Figure 5. I am confused by the x-axis label ‘middle year of decade’ as the temporal resolution seems to show you have a measure for each year?

Response: That is correct. Many studies on coral bleaching projection modelling follow the approach of calculating metrics like ‘high frequency bleaching’ or ‘annual severe bleaching’. These metrics are calculated for each year but are based on the information from the surrounding decade (similar in concept to a running mean with a window size of 10 years, 5 years either side). This is explained in the methods on Lines 528-530. While this is an important detail for the technical aspects of the paper, for the more general audience it is unlikely to be so critical. Therefore, to address your comment we have changed the axis title to say ‘Year’ instead.

Lines 570-573: *For each year, decadal 'high-frequency bleaching' condition was assigned to a coral reef cell if two or more years from the surrounding decade (10-year window centred on the year in question) had experienced mass bleaching conditions.*

Line 204: see major comment on terminology for this, suggest e.g “according to the most parsimonious model” or similar

Response: We have made the suggested change.

Lines 224-225: *the most-likely historic rate of change in thermal tolerance according to the most parsimonious model is 0.1 °C/decade.*

Line 208. Suggest ‘experienced’ rather than ‘faced’

Response: We have made the suggested change.

Lines 230: *coral reefs have experienced over the past decades.*

Line 215. Suggest: ‘Despite levels of heat stress and light intensity that were broadly equivalent to the ‘conditions that led to’ mass bleaching in 1998 and 2010 at the same reefs’

Response: We have made the suggested change.

Lines 237-238: *despite levels of heat stress and light intensity that were broadly equivalent to the conditions that led to mass bleaching in 1998 and 2010 at the same reefs.*

Line 224. Be clear that the mitigation is more under Paris than under 4.5. Also worth mentioning that bleaching likelihood actually starts to decrease again under the Paris towards end of century.

Response: This is a great idea to include here. See the restructure of sentences to reflect this suggestion.

Lines 244-248: *If the historically realised increases in thermal tolerance can continue through the 21st century, then high frequency bleaching could subside after 2050 under the most optimistic Paris Agreement scenario which limits warming below 2 °C (SSP1-2.6) and may be limited substantially even under a middle-of-the-road emissions scenario (SSP2-4.5). However, high frequency bleaching cannot be mitigated only delayed under higher emissions scenarios (SSP3-7.0 and SSP5-8.5), ...*

Line 232. Name some of the other ‘global regions’ here?

Response: We have now mentioned these other regions and given more expanded information on each case study to highlight our point that bleaching susceptibility can change depending on environmental and biological factors.

Lines 257-260: *For example, cloudy weather may have lowered the susceptibility of corals to heat stress in the Society Islands²⁸, and historic exposure to a selective event in Singapore and Malaysia may have led to higher heat stress resistance in some key reef taxa due to adaptation¹².*

Line 252. I think you mean bleaching susceptibility decreased, or bleaching tolerance increased?

Response: We have made the suggested change.

Lines 280-281: *the bleaching susceptibility of coral assemblages decreased in the years post-2016.*

Line 254. Stick with 'taxa' or genera, rather than species. Unless someone monitored at species level, shifts in species composition could have occurred within the Acropora or Pocillopora which would be another potential explanation for increased heat tolerance. I think worth noting this here, see major comment.

Response: Thanks for highlighting this point. We agree entirely and have changed the text to reflect broader coral community composition instead.

Lines 283-284: *Such a lack of profound shifts in coral community composition suggests that...*

Line 257. I am unclear what you mean by 'individual heat tolerance'

Response: Some of the work that our lab has been doing has been aimed at getting a robust measure of heat tolerance for different individual corals with the later aim of using them for selective breeding. Therefore, the individual (rather than the population) is important. However, I admit that it is not clear here why we use this term. To avoid further confusion, we have removed it.

Lines 286: *Evidence suggests that individual heat tolerance is widespread.*

Line 272. Add reference(s) – e.g. Gilmour et al 2013, Recovery of an isolated coral reef system following severe disturbance

Response: We have made the suggested change.

Line 282. 'Understanding the extent to which thermal tolerance can increase naturally on reefs' ... would be good to also work mechanisms into this sentence, i.e. 'and identifying the key mechanisms allowing this increase'

Response: We have made the suggested change.

Lines 312-313: *Understanding the extent to which thermal tolerance can increase naturally on reefs and identifying the key driving mechanisms could...*

Line 286. Suggest: more extreme and 'more frequent'. Also mention chronic increases in ocean temperature?

Response: We have made the suggested change.

Lines 316: *survive progressively more intense and frequent temperature events.*

Line 423. For 'the' warmest

Response: We have made the suggested change.

Lines 475: *For the warmest.*

Line 435. Briefly mention some of these survey methods, e.g. phototransects, point surveys?

Response: We have made the suggested change.

Lines 487-489: *multiple survey methods in the data collection (e.g., photo transects, point intercept transects, video transects), bleaching observations in the database (reported as percentage of corals bleached), are summarised as ...*

Line 438. Suggest: from 0 to 3: 0 = no bleaching (0%), 1 = mild (1-10%), 2 = moderate (11-50%) and 3 = severe bleaching (>50%).

Response: We have made the suggested change.

Lines 490-491: *from 0 to 3: 0 = no bleaching (0%), 1 = mild bleaching (1-10%), 2 = moderate bleaching (11-50%), and 3 = severe bleaching (>50%).*

Line 476. Add a reference, maybe (Glynn and D'Croz, 1990)

https://coralreefwatch.noaa.gov/product/5km/tutorial/crw10a_dhw_product.php

Response: We have made the suggested change.

Lines 534: *represents the theoretical temperature stress threshold for corals⁵⁰.*

Line 497. Need to say what a realisation is

Response: We have made the suggested change.

Lines 556-558: *The lowest number realisation (r1 if available, realisations are climate model runs with different initial conditions) was used as the representative run of each GCM^{20,52}.*

Extended Data Fig1. Could add vertical line at 2010 and refer to Figure 2 in results

Response: Thanks for this suggestion. We have made the relevant changes to the file and have added to the figure caption.

Lines 673-676: *A shaded grey region is show for the year 2010 (inspected in more detail in Fig. 2), showing the levels that thermal tolerance (temperature stress thresholds) would have reached after 22 years of simulated enhancement.*

Extended data Table. Insert 'Summary of GCMs used in table legend rather than 'GCM data' because this table doesn't contain the data.

Response: We have made the suggested change.

Lines 693: *Summary of Global Circulation Models (GCM) data used.*

Reviewer #2 (Remarks to the Author):

This is an important manuscript. There are however a few issues that need clarification, particularly concerning some of the methods of coral bleaching. The authors have provided detailed methods of sea-surface temperature and the simulations and projections, which are all clear, however, they gloss over the major response variable, which is coral bleaching. The highest resolution of SST appears to be 5 km, and we can assume that coral bleaching was observed at reef sites, recorded as geographical coordinates, but there is no mention of how the authors coupled the predictor variable (SST) and the response variable (bleaching).

Response: Thanks very much to reviewer 2 for their helpful feedback and suggestions. To address this comment, we have added extra detail on the method used to pair the bleaching and heat stress datasets here.

Lines 494-497: *For each bleaching record the corresponding value of heat stress for the date of that observation was assigned from the encapsulating 5km grid cell. For records that only reported a survey month and year, the DHW from the 15th day of the month was used.*

How many SST pixels were there for Palau and how many sites were there?

Response: The total number of coral reef SST pixels present for Palau is 152. This has now been added to the manuscript following a previous comment from reviewer 1.

Lines 569-570: Coral reef cells were identified as those that intersect coral reefs in Palau^{56,57} (N=152).

It seems that some SST pixels will contain more than one study site (coordinate), if so what did the authors do to avoid pseudo-replication? Were the coral bleaching values averaged for each pixel?

Response: In our study, the genuine replicate is the bleaching observation, and we were predicting at the level of a site. Although some bleaching records do occur in the same 5km grid cell, they are still independent observations of bleaching severity, and we assume that the temperature experienced in the 5km grid cell was the same for all sites within it. The fact that those bleaching observations that come from the same grid cell will share the same DHW value is a limitation of the spatial resolution of the data as the satellite-based skin temperature is only serving as a proxy for the true temperature environment that a surveyed patch of reef experienced. This is equivalent to saying that the sites within 5km resolution grid cells all suffered the same temperature as a fixed effect. On this basis, we did not average out all bleaching survey records that happened to be in the same CoralTemp grid cell. This is not pseudoreplication, it is simply one or more sites experiencing the same temperature regime. One of the main issues with pseudoreplication in a simple statistical model (e.g., a simple regression), is that the model treats all datapoints as independent observations. However, as we know this is not true for any spatial problem, and that is where the INLA-based statistical modelling approach comes in. INLA spatial fields explicitly model the spatial dependency among events (bleaching observations) in terms of the spatial disposition (separation) of the sites themselves. Specifically, the triangular mesh resolution used in INLA is much finer than the regular grid resolution of CoralTemp, and the mesh specifically quantifies the spatial dependency between sites. Therefore, even if multiple bleaching observations happen to be located in the same CoralTemp grid cell (and thus share the same DHW value), the spatial-correlated error from the INLA model accounts for this. The spatial correlated error does not pertain to the 5km grid of the one of the fixed effects but rather to the variation in the response, and therefore accounts for sub 5km non-independence issues that may arise. To address your concern, we have added to the methodology to reflect this point.

Lines 502-505: this is a map of spatially correlated uncertainty that shows where bleaching is under- or over-predicted for a given heat stress dosage, at the scale of the individual reef sites, not the grid resolution (5km) of the heat stress fixed effect, whilst accounting for non-independence of bleaching observations that happen to occur in the same grid cells.

This is particularly important so that readers can understand Figure 4, the “observations” predicted correctly.

Response: Note, that we have updated the figure caption to describe more clearly what is represented by observations, correct predictions, under-predictions, and over-prediction. See comment above.

Some other concerns.

Abstract. Line two should read more intensive (not intense)

Response: We have made the suggested change and have added the word ‘more’. However, we have not used the term intensive, as we think that the word ‘intense’ is

grammatically correct and more appropriate (dictionary definition of intense = of extreme force, degree, or strength).

Lines 16-17: *corals will need to endure progressively more intense and frequent marine heatwaves.*

Change the 5th sentence to read. This led to less severe bleaching impacts than predicted, indicating...

Response: Here, we had 13 models, each which made different predictions, with one of our models predicting this correctly. However, we refer to the counterfactual of no increase in thermal tolerance, by using the phrase 'than would have been predicted otherwise'. We wanted to make this point here in the abstract, and so have decided not to change this sentence.

High-frequency bleaching (compound adjectives are hyphenated)

Response: We hyphenated the term 'high frequency' made at each mention as suggested.

Throughout the manuscript, compared to should read compared with

Response: Changes have been made at each mention as suggested.

Write in parallel, 3rd to last sentence, "yet can be only delayed..."

Response: We have made the suggested change.

Lines 25: *yet can only be delayed.*

Main text

A few important references are missing showing higher resistance to repeated coral bleaching, including Diane Thompson's work.

Response: Thanks for highlighting this work. As suggested, we have included citation of Diane Thompson's paper: Corals escape bleaching in regions that recently and historically experienced frequent thermal stress (**Line 45**).

The last sentence of the first page should read: "Such increases in thermal tolerance can come about naturally through various mechanisms, including turnover, genetic adaptation, and acclimatisation." In that way, the authors do not need to awkwardly enumerate the following paragraph.

Response: We have made the suggested change.

Lines 46-47: *Such increases in thermal tolerance can come about naturally through various mechanisms, including community composition turnover, genetic adaptation, and acclimatisation.*

The second page, 3rd paragraph.

Should read: "This is achieved by simulating..."

Response: We have made the suggested change.

Lines 78: *by simulating 13 different.*

Page 4.

This section needs clarification as it is confusing which data were actual measurements and which data were simulated. I assume the 2010 SST data were actual measurements. As it is written, it appears that the 2010 marine heatwaves were simulations.

Should read: “Under a simulated thermal tolerance enhancement...”

Response: We have cleaned up a few of the sentences to clarify these points.

Lines 118-125: *Despite the same measured SST history, the DHWs profiles corresponding to each simulation were strikingly different, particularly in the later years after there had been more time for thermal tolerance to shift from baseline temperature stress threshold level. Giving the 2010 marine heatwave as an example (Fig. 2, showing DHW based on average SST of all Palauan reef), the maximum DHW reached without any thermal tolerance enhancement was 6.6 °C-weeks (Fig. 2, yellow line). Whereas under a simulated thermal tolerance enhancement of 0.1 or 0.2 °C/decade, the maximum DHW reached was 1.7 and 0.3 °C-weeks, respectively.*

Figure 2. The actual data measurements and simulations need to be clarified here as well. Figure 2, should not be called “Example mean...”, simply “Mean sea surface temperature...”

Response: See new figure caption as per suggestion of reviewer 1 (comment above).

The historically realized increase in thermal tolerance section. The sentence on spatial correlation needs to be rewritten. How were they accounted for?

Response: We have addressed this by including some more information about the estimating the spatial random effect to lead readers toward our more detailed methodology section where this is expanded in full.

Lines 146-150: *Spatial correlation in model residuals would breach the ‘independent observations’ assumption of simple logistic regressions but were accounted for here by explicitly estimating spatial variations in model uncertainty using a Bayesian approach. This accounted for under-prediction of bleaching observations in the north of Palau compared with the south (Extended Data Fig. 3).*

Figure 3 caption. Clarify. And model parsimony was evaluated using DIC

Response: We have made the suggested changes, and again highlight the ‘simulated’ thermal tolerance enhancement rate, as per the previous comment.

Lines 155-158: *and model parsimony was evaluated using the deviance information criterion (DIC). Notably, the most parsimonious predictions with lowest DIC were achieved for a simulated thermal tolerance enhancement rate of 0.1 °C/decade.*

Figures 4B and C. Need to clarify how the response variable was estimated (see opening comment).

Response: See new section added to the figure caption and reply to a similar comment left by reviewer 1 above.

Discussion. Paragraph 3. The bleaching susceptibility should read “the bleaching susceptibility of coral assemblages decreased” because if sensitive taxa are lost the system becomes more tolerant, not more susceptible.

Response: Thanks for highlighting this. We have made the relevant changes.

Paragraph 4. The authors are assuming that there were residual populations.

Response: The long-term recovery paper in Proc B (Gouezo et al) shows that by 2002 there was some Acropora and Pocillopora at certain sites. As such it is likely that there was a residual Acropora population, albeit small. This is already alluded to in the paragraph where we have written: ‘thereby increasing the frequency of heat tolerance genes in the remaining population’.

Paragraph 6. Delete the first sentence. Rewrite the awkward final sentence of this paragraph.

Response: Thanks for these suggestions. We have rephrased the end of the paragraph, however we reduced the first sentence rather than removing it entirely, as it sets the scene for the second sentence in the paragraph.

Lines 316-317: *Throughout the coming century, corals will need to survive progressively more extreme and frequent temperature events.*

Lines 321-326: *Thus, while we know that tolerance is increasing on decadal time scales, it remains a priority to study the diversity of potential mechanisms driving these trends. While our study demonstrates the existence of an innate ecological resilience to climate change, this is insufficient to mitigate severe impacts under middle-to-high emissions scenarios, highlighting the continued need to reduce carbon emissions and to fulfil Paris Agreement commitments.*

Methods. Bleaching prediction models. Delete “provides a more intuitive” (more intuitive than what? A frequentist model, yes, but that is unclear), simply write: “...statistical approach to spatially explicit...”

Response: To address your concern, here we simply remove the comparison in this sentence by removing the word ‘more’.

Lines 498-499: *This Bayesian statistical approach provides an intuitive spatially explicit estimation of model uncertainty.*

For the equation, remove GMRF, and replace it with N for the spatial mu component.

Response: In the linear equation: $\text{logit}(\pi_i) = \beta_0 + \beta_1 \times \text{DHW}_i + \mu_i + \varepsilon_i$, the first error term (μ_i) is for modelling the uncertainty that is due to spatial autocorrelation, and the second error term (ε_i) is the typical normally distributed error that is not spatially correlated. Since the Gaussian Markov Random Field (μ_i) is not a simple normal distribution (ε_i), here we retain the notation of GMRF(...).

REVIEWERS' COMMENTS

Reviewer #1 (Remarks to the Author):

I am satisfied that my concerns and suggestions for improvement from my first review have now been addressed and the manuscript clarified and improved as a result. I am happy for the manuscript to proceed to publication.

I look forward to seeing it in print.

Reviewer #2 (Remarks to the Author):

The authors have addressed all of my comments and concerns and the manuscript has greatly improved. It was useful to emphasize the strengths of the Bayesian analysis and highlight the analytical strengths of the spatially correlated error terms because this manuscript is essentially a modeling exercise, although an excellent one, and the readers of the work need to be convinced that there are no glaring weaknesses with the approach.

They could expand their response slightly to mention, in the manuscript, what the grid does and how spatial dependency of bleaching is adjusted for in the analysis.

One annoying aspect remains, however, in that the authors self-cite a minor study on the solitary islands, on one coral species. Lachs et al. 2021, should not be used at the beginning of the manuscript to describe marine heatwaves and global bleaching. Best to cite some of the giants in the field, such as Peter Glynn or Barbara Brown, here instead.

We would like to thank the two anonymous reviewers for your interest in our work, and for helpful and constructive comments. In this response to reviewers document, please note that all author responses are shown in **blue**. Quotes from the manuscript are shown in *italics*. Line numbers are shown for the simple view of the revised manuscript (tracked changes not showing), and revisions/additions compared to the last draft are shown as *italics*.

Reviewer #1 (Remarks to the Author):

I am satisfied that my concerns and suggestions for improvement from my first review have now been addressed and the manuscript clarified and improved as a result. I am happy for the manuscript to proceed to publication.

I look forward to seeing it in print.

Response: Thank you for your helpful comments.

Reviewer #2 (Remarks to the Author):

The authors have addressed all of my comments and concerns and the manuscript has greatly improved. It was useful to emphasize the strengths of the Bayesian analysis and highlight the analytical strengths of the spatially correlated error terms because this manuscript is essentially a modeling exercise, although an excellent one, and the readers of the work need to be convinced that there are no glaring weaknesses with the approach. They could expand their response slightly to mention, in the manuscript, what the grid does and how spatial dependency of bleaching is adjusted for in the analysis.

Response: Thank you for your helpful comments on this which prompted to explain the spatial modelling approach more clearly in the main manuscript.

Line 150-153: *In essence, non-independence of nearby observations are accounted for by quantifying spatially correlated error in bleaching predictions and using this to adjust the error term of model, usually increasing uncertainty toward a more realistic level compared to an equivalent non-spatial model.*

One annoying aspect remains, however, in that the authors self-cite a minor study on the solitary islands, on one coral species. Lachs et al. 2021, should not be used at the beginning of the manuscript to describe marine heatwaves and global bleaching. Best to cite some of the giants in the field, such as Peter Glynn or Barbara Brown, here instead.

Response: We have changed this citation to Glynn, P. W. Coral reef bleaching: facts, hypotheses and implications. Glob. Chang. Biol. 2, 495-509 (1996).